

# Detecting Wave Features in Doppler Radial Velocity Radar Observations

Matthew A. Miller[1], Sandra E. Yuter[1], Nicole P. Hoban[1], Laura M. Tomkins[1], and Brian A. Colle[2]

[1]North Carolina State University, Raleigh, NC
[2]Stony Brook University, Stony Brook, NY

**Correspondence:** Matthew A. Miller (mamille4@ncsu.edu)

**Abstract.** Mesoscale, wave-like perturbations in horizontal air motions in the troposphere (velocity waves) are associated with vertical velocity, temperature, and pressure perturbations that can initiate or enhance precipitation within clouds. The ability to detect velocity waves from horizontal wind information is an important tool for atmospheric research and weather forecasting. This paper presents a method to routinely detect velocity waves using Doppler radial velocity data from a scanning weather

radar. The method utilizes the difference field between consecutive PPI scans at a given elevation angle. Using the difference between fields a few minutes apart highlights small scale perturbations associated with waves because the larger scale wind field changes more slowly. Image filtering retains larger contiguous velocity bands and discards noise. Wave detection scales are limited by the size of the temporal difference relative to the wave motion and the radar resolution volume size.

# 1   Introduction

Mesoscale velocity waves are common in the atmosphere (Holton and Hakim, 2013). Both gravity waves and Kelvin-Helmholtz waves frequently occur and are associated with clouds and precipitation. Gravity waves (also called buoyancy waves) arise when an air parcel is displaced upwards or downwards within a stable layer and buoyancy causes an oscillation. Kelvin-Helmholtz waves can develop when there is a shear-driven instability. Tropospheric gravity waves can generate and enhance

precipitation within orographic clouds, deep convective storms, and drizzling clouds (e.g. Mapes, 1993; Gaffin et al., 2003; Fovell et al., 2006; Parker, 2008; Allen et al., 2013). Gravity waves can be triggered by several mechanisms, including latent heat release, unbalanced flow within a jet stream, and mountainous terrain (Mapes, 1993; Koch and O'Handley, 1997). Gaffin et al. (2003) described how mountain gravity waves initiated by downslope flow near the crest yielded a banded heavy snow event over the southern Appalachian region. Allen et al. (2013) found that gravity waves triggered by geostrophic readjustment

of the subtropical jet stream yielded transient lines of drizzle in a region of marine stratocumulus clouds. Fovell et al. (2006) demonstrated how gravity waves caused by latent heat release inside a squall line caused precipitation to form ahead of the squall line (action at a distance). Several studies have used radar data to infer microphysics perturbations co-located with





Kelvin-Helmholtz velocity wave signatures, including along a cold-frontal zone (Houser and Bluestein, 2011), along a warm-frontal zone and above the boundary layer (Rauber et al., 2017), and in a winter storm within a stable layer over a peak in

terrain (Geerts and Miao, 2009). Whereas Kelvin-Helmholtz waves tend to move with the prevailing wind, gravity waves can move perpendicular to or against the prevailing wind or can be geographically fixed relative to mountainous topography.

Snow storms impacting the northeastern US often have precipitation organized into mesoscale bands within the northwest and northeast quadrants of the cyclone (Ganetis et al., 2018). Precipitation accumulation is sensitive to the occurrence, intensity, and propagation of these bands (Novak et al., 2004, 2008; Novak and Colle, 2012; Ganetis et al., 2018; Kenyon et al., 2019).

Investigation of Doppler radial velocity data for northeastern US snow storms has shown the frequent occurrence of transient, banded velocity perturbations that move perpendicular to the mean flow in a wave-like pattern. These velocity perturbations often co-occur with mesoscale snow bands (Hoban et al., 2017). Figure 1 illustrates a snow storm on 12 January 2011 in the Boston, MA area in which WSR-88D radar data for the 0.5° elevation scan shows sets of roughly linear, southwest to northeast oriented, localized reflectivity enhancements (bands) associated with linear perturbations in radial velocity (waves).

This paper describes a method to routinely detect waves by their perturbations of the horizontal wind using Doppler radial velocity radar observations. We refer to these features in the radial velocity data using the generic term "velocity waves" because identification of specific wave types often requires contextual information in addition to radial velocity. This method can detect gravity and Kelvin-Helmholtz waves as well as other phenomena that produce roughly linearly oriented horizontal velocity perturbations with an amplitude and on a spatial scale resolvable by Doppler radar, such as gust fronts, hurricane rain

band convergence, sea breezes, and terrain-flow blocking. Additional analysis can be done to determine if a velocity wave is in fact a propagating gravity wave, which has a quadrature relationship (90° shift) between the vertical motion and the horizontal wind/pressure perturbation.

Radar has been used to detect and observe the structure of velocity waves in the atmosphere for decades. For example, Ottersten et al. (1973) describe clear-air radar observations of gravity waves in the troposphere. Stober et al. (2013) used

Doppler radial velocity observation to detect and measure the properties of gravity waves in the upper atmosphere from 75 to 100 km altitude. They estimated and then subtracted the mean horizontal wind velocities from their radial velocity observations of the mesosphere. The residual velocities depicted a monochromatic gravity wave that was analyzed by hand in order to estimate its wavelength, phase velocity, and propagation direction. These studies and many like them rely on using clear-air radar reflectivity data where backscattering particles act as passive tracers. These clear-air reflectivity-based methods require

high radar sensitivity and will not work when precipitation—which by definition has a fall speed—is present.

## 2   Methods

The technique described in this paper is designed for use in the lower troposphere and works well even if the background wind field is complex. By subtracting two consecutively scanned Doppler radial velocity fields, we can remove the more slowly varying background wind and isolate relatively faster moving velocity wave features that have noticeable movement between

consecutive scans.





The WSR-88D Level II Doppler radial velocity data are first processed to unfold (dealias) velocities (Helmus and Collis, 2016). To improve the results of the automated dealiasing, radial velocity values are removed where reflectivity is less than 0 dBZ and small speckles are removed. Two sequential polar coordinate Doppler radial velocity fields from the same elevation angle are subtracted to obtain a difference field. For RHI scans, the same azimuth angles should be used. Assuming that the background velocity field is relatively unchanged between successive sample volumes, taking the difference between successive radial velocity fields shows where features in the radial velocity data have propagated in space.

We illustrate the wave detection method using data from a snow storm on 26 December 2010 observed by the US National Weather Service (NWS) radar in Upton, NY (Fig. 2). The storm's low pressure center is 200 km to the southeast of the radar and is moving toward the northeast. The near surface winds are northerly. In the southeast quadrant, the wind direction turns clockwise with height. Figure 2a and Figure 2b represent two consecutive dealiased radial velocity observations in polar coordinates, and Fig. 2c is the difference between the two fields (*time 2 - time 1*). When velocity waves are present, the difference field contains banded features with both negative and positive values that represent a temporal change in the radial velocity field. The velocity signatures of interest manifest as perturbations from the background velocity field shaped as long thin lines. Sets of waves present as sets of roughly parallel lines and resemble plane waves. For waves moving toward a radar, positive values of radial velocity acceleration (i.e. *time 2* radial velocity > *time 1* radial velocity) represent areas of horizontal divergence, while negative values represent areas of deceleration and associated horizontal convergence.

To provide better visual distinction between nearby velocity bands, the difference field was converted to a binary field while retaining the native polar coordinates. We chose to use the negative portion of the radial velocity difference field in order to focus on upstream convergence and inferred upward motion. Since we applied the wave detection to the 0.5° elevation angle PPI, the associated velocity waves are detected at lower altitudes of the storm. Because the radar only observes the radial component of the total wind vector, convergence is implied but not guaranteed because the unobserved component of the wind may compensate for radial convergence along the radar beam. The portion of the difference field with radial velocities less than a threshold value of -1 m s$^{-1}$ was set to one and the remaining portion of the field was set to zero (Fig. 2d). Removal of areas with velocity perturbations near zero reduces noise. Isolating higher amplitude velocity waves marked by higher velocity perturbations is not very sensitive to the specific threshold value used. For WSR-88D data, we found that using -1 m s$^{-1}$ as a binary threshold value balances removing noise and detecting the features of interest. The binary threshold value appropriate for any specific data set will vary based on characteristics such as the phenomena of interest, the sampling frequency of the radar, and the noise level of the radar.

For the next step, the polar coordinate binary field was interpolated to a 0.5 km Cartesian grid using a nearest neighbor algorithm. A grid resolution of 0.5 km is sufficient for Level II WSR-88D data based on the 0.93° beamwidth and the unambiguous range (137 km in this example). The goal is to oversample the data to preserve the detail native to the original polar field. As a last step, we filtered out objects smaller than 16 km$^2$. We then used the skimage morphology Python package (van der Walt et al., 2014) to remove eight-connected, continuously connected areas of positive wave detection smaller than 16 km$^2$. In implementing a minimum size filter, the users are making a choice about the size scale of waves they are interested in. Waves in storms are common across a variety of spatial scales and orientations. Myriad waves often overlap. A given minimum or



maximum value for a size filter will emphasize the detection of a certain scale of waves. The appropriate Cartesian grid resolution and area size to filter out is a function of the radar characteristics and intended application. In this example, we focused on the larger velocity waves and filtered out the smaller scale features including most of what appears to be spatial noise in the western half of the domain.

The resulting field (Fig. 2e and associated animation in the Video Supplement) contains several sets of parallel velocity wave features. The most prominent are the set with their long axis aligned from SSW to NNE. The horizontal wavelength of these waves is on the order of 12 to 18 km. Examination of sequences of detected waves shows the wave train propagating toward the NW and spanning the radar domain. Embedded among the longer SSW to NNE waves are collections of shorter wavelength horizontal waves oriented in a variety of directions. In the southeastern quadrant of the radar domain, the detected velocity

features are less spatially coherent in part because the radar echo is more sporadic in that region.

Within the limitations described below in Section 2.1, we can use this method of taking the difference between successive Doppler radial velocity fields and converting to a binary field to identify velocity wave structures. Examining the binary difference fields as they evolve gives information on the speed and direction wave trains propagate. Measuring the distance between adjacent wave features gives the horizontal wavelength. Examining several different elevation angles and/or computing

a vertical cross section based on multiple elevation angles provides information on the spatial coherency of the waves with height and the depth of the waves. The velocity difference technique described here can also be applied to sequential Doppler velocity data from RHIs scanned along the same azimuth to detect possible waves (see Section 3.3).

Tracking coherent features in the wave detection data across successive volumes will allow the user to calculate the wave propagation direction and speed by measuring the direction and distance of the displacement of these features and accounting

for the difference in sampling time. There are several different approaches to performing such an analysis using manual and automated methods. The details of any successful approach will be largely dependent on the specifics of the radar data, the features of interest, and a user's applications. In some situations, the application of non-rigid image registration tools as well as optical flow and related methods for feature tracking have shown promise.

## 2.1 Limitations

### 115 2.1.1 Detection

The ability to resolve the velocity waves in Doppler radial velocity data is constrained by the spatial resolution of the radar, the temporal sampling interval between consecutive scans, and the temporal scale of changes in the background wind field. Additionally, there is the intrinsic limitation of radial velocity data in that the radar only measures the component of the wind along azimuths originating at the radar. The WSR-88D radar can detect velocity changes $\geq 0.5$ m s$^{-1}$ (NOAA, 2017). The

size of radar sample volumes is a function of the radar antenna beamwidth, the size of the range gates (pulse length), and the range from the radar (Battan, 1960). At minimum, the velocity waves must propagate horizontally at least the equivalent of one radar resolution volume in distance along the direction of wave motion to show as a difference in the PPI radial velocity data. In practice, we have found that the waves need to propagate at least 3 times the horizontal radar resolution size to reduce





uncertainty related to beam filling and velocity aliasing and to have a signal clearly distinguishable from the background noise.

For the US WSR-88D network, the radars have a typical half-power beamwidth of 0.93° and the length of the range gates for Doppler velocity are 250 m (NOAA, 2006). At a range of 100 km, the WSR-88D radar resolution volume size is an approximate cylinder 1.6 km wide by 250 m long. The temporal sampling interval varies among different volume coverage patterns (VCPs). For the US WSR-88D network VCPs, the time between consecutive scans at the same elevation angle is typically between a few minutes for scans designed for severe weather and up to approximately 10 minutes for scans designed for clear air and light

rain or snow precipitation (Rauber and Nesbitt, 2018). For a radar resolution volume horizontal distance of 2000 m and a time interval between consecutive scans of 300 seconds, the velocity wave needs to propagate at a minimum of speed of 6.67 m s$^{-1}$ to traverse a sample volume.

For this wave detection method to work best, the background wind field must change as little as possible between the radial velocity observations used to calculate the difference field. Turbulence must also be low since this creates noise in the radial

velocity difference fields. Time scales for storm evolution usually increase as storm spatial scales increase. Large, synoptically driven systems typically evolve more slowly than rapidly changing, convective-scale thunderstorms. If the spatial scale of the background wind change is small, it creates the equivalent of small scale noise that can be mitigated by techniques such as removing contiguous areas in the binary difference field that are small in area and not likely to be waves. If the changes in the background wind field are large in spatial scale, they will mask the presence of velocity waves.

Not every linear velocity band feature detected by this method will correspond to a wave. Wind shifts along lines, including convergence lines, gust fronts, and fronts, will be detected as single velocity bands and will usually move with the prevailing wind. In contrast, propagating wave features will appear as sets of roughly parallel linear velocity bands that move in concert and may or may not move with the same speed or direction as the prevailing wind. Users should keep in mind the meteorological context of the observations so they can make informed judgements about which velocity band features are waves and which

are not.

### 2.1.2  Wave motion

As compared to simple wave detection, there are more stringent conditions to obtain an accurate speed and direction for propagating waves. To obtain the correct wave speed, the wave train must move less than half its wavelength between sample times. A wave train that moves more than half its wavelength can appear to propagate in the wrong direction. A wave train that

moves exactly its wavelength in a sample period will appear stationary. The sample time, in seconds, must be less than

$$t < \frac{\lambda}{2 \times V} \tag{1}$$

where $t$ is the sample time in seconds, $\lambda$ is the wavelength in meters, and $V$ is the wave propagation speed in meters per second. An NWS radar in VCP-12 completes a volume approximately every 240 s. A wave with a wavelength of 10 km would therefore have to have a propagation speed of less than 20.8 m s$^{-1}$ to avoid errors in estimating propagation speed and direction. Further

discussion of this issue is in Section 3.1.





## 3    Application to Idealized Waves and Storm Examples

In order to test and illustrate the velocity wave detection method in a controlled environment, we wrote software to create idealized plane waves and sample them as a radar would (Miller, 2021). This allowed us to verify that we are able to correctly estimate wavelengths, depths, speeds, and directions without the uncertainty and noise found in real-world data. In the idealized

examples below, radial velocity data were generated for plane waves with various defined wavelengths, depths, orientations, and amplitudes of the horizontal velocity perturbations. The code was used to compute the wave characteristics in a Cartesian coordinate system as well as to sample the waves along elevation angle PPIs for a set of user-specified radar characteristics, including maximum unambiguous range, number and size of range gates, and beamwidth. Standard refraction and earth curvature were accounted for using the 4/3rds Earth approximation (Doviak and Zrnić, 1993; Rinehart, 2010). All simulated radar

points in a PPI were sampled all at the same time as compared to ray-by-ray sequential sampling as a real radar would scan.

### 3.1    Single Wave Example

We constructed an idealized plane wave by prescribing horizontal velocity perturbations as a sinusoidal plane wave. Figure 3 and Animation-Figure3 show vertical cross sections of horizontal velocity taken perpendicular to the long axis of example idealized waves (upright and tilted) and a background wind field of 0 m s$^{-1}$. In these examples, wavelength was set to 30.7 km,

wave depth to 4500 m, and the maximum horizontal velocity perturbation is 4 m s$^{-1}$. The figure also illustrates the horizontal pattern of calculated radial velocity as a weather radar would observe them for a 0.5° elevation PPI, maximum range of 150 km, and 500 m range gates. The waves are moving to the west (270°) (Animation-Figure3). As expected, the radial velocity values are zero in the north to south direction where the beam is perpendicular to the direction of the wave's velocity perturbations. When the waves are tilted (Fig. 3bd), they appear bowed in PPI radar images. This effect is caused by the height of the radar

beam over the surface increasing as a function of range.

When the time between samples is too long (Eq. 1), the apparent wave speed will be wrong. Video Supplement movies Animation-MotionStudy1 and Animation-MotionStudy2 illustrate mischaracterizations of wave speed when the temporal sampling is insufficient. As described in Section 2.1.2, if the temporal sampling is equal to the time a wave takes to travel an integer multiple of its wavelength, the wave will appear to not be moving. A wave could also appear to be propagating the

incorrect direction if the temporal sampling frequency is too low. The animations feature a single wave with a wavelength of 15 km moving toward 305° with a variable propagation speed and a constant simulated radar sampling frequency 250 s. In Animation-MotionStudy1, the wave propagation speed of 30 m s$^{-1}$ and radar sampling frequency of 250 s are such that the wave moves half its wavelength each sampling interval. This results in the peaks and troughs of the waves alternating positions at successive sample times and yields a flashing animation where no wave motion is discernible. At double the wave speed or

half the sampling interval the wave would travel a full wavelength and appear to be stationary in an animation (not shown). In Animation-MotionStudy2, the wave propagates at 40 m s$^{-1}$ and moves more than half its wavelength between samples but less than its full wavelength. This results is the appearance that the wave is moving in the opposite direction at a slow speed. These effects can also be illustrated by using a fixed wave propagation speed and a variable radar sampling frequency.




### 3.2 Two Waves Example

In real storms, several waves with different wavelengths, orientations, and depths often coexist. Figure 4 and Video Supplement Animation-Figure4 show PPIs of two different idealized waves superimposed. The simulated radar configuration and temporal sampling are the same as in Fig. 3. The two waves were generated separately and the velocity fields were then summed. In this example, the longer wavelength is 25 km and the shorter wavelength is 8 km. The 25 km wave is oriented toward 290°, extends from 0 to 4500 m altitude, and has a maximum horizontal velocity perturbation of 4 m s$^{-1}$. The 8 km wavelength wave is oriented toward 250°, extends from 0 to 1000 m altitude, and a maximum horizontal velocity perturbation is 2 m s$^{-1}$. Within the layer between 0 to 1000 m, altitude both waves are superimposed.

In the 0.5° elevation PPI (Fig. 4), the beam tops the 1000 m wave at about 70 km range. In cases where one is certain the beam is overshooting a wave train, the range at which this occurs can be used to infer the depth of the wave by referencing how the radar beam height increases as a function of range. When two waves with different wavelengths overlap, the longer wavelength wave typically dominates and can obscure the signal from the smaller wavelength plane wave. The obscuring of the shorter wavelength waves is most evident when the orientation of the two wave sets are similar. As the number of superimposed waves increase, the more the wave detection information degrades to noise.

The contrasting wave characteristics in vertical cross section are illustrated in cross sections of the idealized wind field and in the RHIs constructed from the PPI scans in Fig. 4 and Animation-Figure4. The RHIs were constructed as one would from operational radar data, by finding the subset of data along a given azimuth in each elevation angle's PPI in the radar volume scan, defining the radar resolution volume center locations corresponding to slant range coordinates, and determining the range-dependent radar resolution volume sizes. The constructed RHIs are along an azimuth of 90°, which is between the two idealized wave orientations of 290° and 250° azimuth and coincides with the cross sections also shown. The set of elevation angles, the beamwidth, and the gate spacing were chosen to match the NWS VCP-12 (Rauber and Nesbitt, 2018) VCP. The tops of the two prescribed waves are denoted in the RHI panels by dashed, gray lines.

The wave detection technique is capable of accurately estimating the tops of wave trains where they are sufficiently resolved. When using RHIs constructed from sets of PPIs, the wave top estimates are most accurate close to the radar where there is greater effective vertical resolution than at farther ranges. The 1000 m deep wave is more difficult to discern because of the overlap between the two waves in the 0 to 1000 m layer. Examination of sequences of cross sections can help mitigate the visualization problem if the waves do not move in phase with each other. Scanned RHIs, when the radar antenna varies elevation angle while holding a given azimuth, typically have higher vertical resolution than RHIs constructed from PPIs and better enable the resolving of wave features.

### 3.3 Winter Storm Examples

We applied the wave detection method to PPI and RHI radar data from a winter storm on 1 February 2021 that impacted the northeast US. Figure 5 shows radar data from 09:03 UTC. The data are from the NWS KOKX radar located on eastern Long Island in Upton, NY. To the north of the radar, there is widespread snowfall and no clear snow banding structure in the





reflectivity data. South of the KOKX radar over the Atlantic Ocean, the dominant precipitation type is rain. Within the rain region and to the southwest of the radar, rain cells are organized into a quasi-linear feature that moves toward the northeast. Across much of the KOKX domain, there are NW to SE oriented velocity bands. Some of these bands are in the rain region and

some in the snow region. Animation-Figure-5 shows that the detected wave-like features move in sync with the rain structures and are likely a signature of convergence associated with the quasi-linear mesoscale organization.

Stony Brook University's KASPR research radar (Kollias et al., 2020; Kumjian et al., 2020) obtained very high spatial resolution radar data during the same storm and observed the fine scale vertical structure of velocity bands (Fig. 6). The high spatial resolution RHIs are almost complete 180° scans from horizon to horizon along the azimuths 179°/359°. The 30 km

range cross section extends from the barrier island along the southern coast of Long Island, northward across Long Island and the Long Island Sound to the southern coast of Connecticut. Examination of linear depolarization data (not shown) indicates that the precipitation within the RHI is all snow.

The KASPR radar has a narrow beamwidth (0.32°) and less than 200 m range resolution, which yields higher spatial resolution than a WSR-88D radar. Prior to wave detection, the KASPR fields were linearly interpolated to a grid with 10 m grid

spacing. For this storm, KASPR RHI scans alternated from north-to-south to south-to-north. Because of this, the waves were calculated using the difference from every other scan so that the temporal spacing between elevations in the scans remain consistent. We used a binary threshold value of -0.4 m s$^{-1}$ and a minimum area filter of 0.03 km$^2$ threshold. The wave detection product derived from the high spatial resolution RHI data shows the presence of vertically oriented, coherent, linear wave patterns between 8 and 16 km south from the radar. Examination of the radial velocity difference field suggests the presence of

two different modes of waves.

Evident in the radial velocity difference data is the presence of waves at approximately 2.5 to 3 km altitude and -5 to -15 km range from KASPR. These waves manifest as perturbations alternating between -1.5 and 1.5 m s$^{-1}$. Spectrum width data show the presence of a shear layer at the same altitude and range, which suggests that the waves at 3 km altitude are Kelvin-Helmholtz waves.

There is also a more subtle set of waves located in the velocity difference field between the bottom of the echo and 4 km altitude and the -12 to -18 km range, with values alternating between -0.5 and 0.5 m s$^{-1}$. These wave features appear to be vertically oriented gravity waves with a wavelength between 1.2 and 2.5 km and with a depth of at least 3 km. The radar is not able to confirm the lower extent of the waves due to beam overshoot of the surface and low-level blocking. By examining successive wave detection fields, the propagation speed component of these deeper waves along the KASPR RHI azimuth is

estimated to be 11.5 m s$^{-1}$ (Animation-Figure-6).

The characteristics of the velocity waves in the 1 February 2021 storm have similar properties to the idealized synthetic waves illustrated in Section 3.2. In the scanned RHIs, examination of the velocity difference fields instead of the simplified binary detection field makes the inferred coexistence of the two different wave modes easier to diagnose. Examination of other variables such as spectrum width further informs the interpretation of the wave detection and radial velocity difference data by

providing context on the turbulence conditions within altitude layers where the waves are present.





The wave detection method can also be applied to RHIs constructed from PPI scans from WSR-88D data and data from similar radars. Figure 7 depicts reflectivity, radial velocity, the difference of radial velocity for successive scans, and the wave detection field for data from the KBOX WSR-88D radar near Boston, MA at 08:45 UTC on 17 December 2020. The RHIs are constructed from PPI data on the 200° azimuth. This azimuth crosses several mesoscale bands of locally increased reflectivity. The tilt of the bands toward the radar with increasing height is consistent with the vertical wind shear observed in the radial velocity data. The radial velocity difference data and the wave detection data derived from that field show at least three wave features that also tilt toward the radar with increased altitude. The wavelength is approximately 20 km, and the observed depth of these wave is at least 3 km. The lack of beam coverage near the surface hampers a more precise estimation. These waves propagate toward the radar from the south between 15 and 20 m s$^{-1}$ (Animation-Figure-7).

### 3.4 Non-Wave Example

Contrasting the structures observed in the winter storm examples, data from Hurricane Sandy on 30 October 2012 (Fig. 8 and Animation-Figure-8) illustrate a storm where the velocity bands and reflectivity bands are usually locked together. There are large, azimuthally oriented velocity bands associated with larger rain bands that are especially evident across New Jersey, southeastern New York, eastern Pennsylvania, and Connecticut. Most velocity perturbations classified by the algorithm are small or amorphous, do not organize into sets of bands, and move in phase with localized, cellular precipitation maxima shown in the reflectivity data. There are linear bands that organize in quasi-parallel sets, but these bands move in concert with precipitation bands around the storm's low pressure center. We interpret the velocity perturbation signatures detected by the method as bands of horizontal wind convergence associated with the hurricane rain bands and not gravity waves. The velocity structures revealed by the wave detection method propagate mainly along their long axis which is not consistent with motion of the synthetic wave data examined in 3.2.

### 4 Conclusions

Our method identifies translating bands of wind perturbations (velocity waves) by subtracting successive radial velocity fields. This method can be used to detect the presence of several types of velocity waves and serve as the foundation for analysis on the role of gravity and Kelvin-Helmholtz waves in storms. Manual or automated measurement of the flagged wave features can be used to determine their wavelength. Tracking the waves in time allows the estimation of the wave propagation direction and speed. Such estimates must be done with care to ensure the spatial and temporal resolution is sufficient to resolve the features of interest and to capture their motion with fidelity (Section 2.1).

The wave detection method was tested using sets of idealized planar waves and accurately captures the idealized waves' characteristics within spatial and temporal resolution limits. Examination of winter storms impacting the northeast US reveals the presence of overlapping packets of quasi-planer, monochromatic waves with wavelengths spanning 1 to 40 km. Radar PPI-derived cross sections of the waves suggest they have depths of several kilometers. Within the same storm, some parallel sets of waves move with the prevailing wind and others move independently of the prevailing wind. Some sets of waves coincide



with banded features of higher reflectivity while other sets move independently of mesoscale reflectivity bands. The nature of interactions among radar-detected velocity waves of several types and reflectivity bands is a topic of ongoing work.

The wave detection output requires informed interpretation because waves are not the only source of linear velocity perturbations. Outflow boundaries, sea-breeze fronts, frontal boundaries, etc., can produce a propagating change in velocity that will be flagged by our method. The user must assess the flagged locations to ensure that they possess characteristics consistent with waves such as presenting as a set of parallel banded features. A creative user can apply various filtering techniques to the data to remove areas that are not likely to be waves. A full discussion of potential approaches is beyond the scope of this paper

because it is highly application dependent.

*Code and data availability.*   MATLAB Code for Idealized Wave Examples

The matlab-RadarSim library (Miller, 2021) used to create the idealized wave figures and animations is available on GitHub. URL: https://github.com/millercommamatt/matlab-RadarSim/releases/tag/v1.0.0 DOI: https://doi.org/10.5281/zenodo.5247995

*Video supplement.*   List of animations with captions and filenames

All animations can be viewed at: https://av.tib.eu/series/1106 Individual animations can be viewed by following the DOI URL.

Animation-Figure-1: Animated plot of bands of locally enhanced reflectivity and coincident velocity waves from Boston, MA WSR-88D radar (KBOX) on 12 January 2011 at 11:00 UTC. Data shown are 0.5° elevation angle PPIs of radar reflectivity, dealiased radial velocity, binary version of the difference field between consecutive radial velocity fields all in polar coordi-

nates, and Cartesian coordinate (0.5 km grid spacing) filtered version of the binary field with eight-connected areas $< 16$ km$^2$ removed. (goes with Fig. 1). Title: KBOX 20110112 Reflectivity, Velocity, and Waves Filename: 20110212_KBOX_4panel.mp4 DOI: https://doi.org/10.5446/54298

Animation-Figure-2: Animated plot of step by step wave detection method illustrated using sequential PPI scans of radial velocity from NWS KOKX radar in Upton, NY on 26 December 2010. Polar coordinate radial velocity fields at 0.5° eleva-

tion angle for consecutive scans at 23:40:00 UTC and 23:45:47 UTC, the difference field of the two radial velocity fields showing both positive and negative temporal velocity changes, the binary version of the negative portion of the difference field from, and the Cartesian coordinate (0.5 km grid spacing) filtered version of the binary field with with eight-connected areas $< 16$ km$^2$ removed (goes with Fig. 2). Title: KOKX 20101226 Velocity and Waves Filename: KOKX_5panel.mp4 DOI: https://doi.org/10.5446/54362

Animation-Figure-3: Animated plot of cross sections and simulated radial velocity images of vertical and tilted idealized velocity waves. Both velocity waves have a wavelength of 30.7 km, a depth of 4500 m, and a maximum horizontal velocity perturbation of 4 m s$^{-1}$. The waves in the left hand panels are upright and the waves in right hand panels are tilted. The upper





plots depict cross sections of the horizontal (zonal) velocity perturbations. The lower plots depict simulate radial velocity observations as would be seen by a NWS WSR-88D precipitation radar with a PPI beam elevation of 0.5°, a maximum range

of 150 km, a 0.95° beamwidth, and 500 m range gates. The solid black lines denote the span of the cross sections shown in the upper and middle panels. The vertical and tilted waves have the same cross-band widths (goes with Fig. 3). Title: Idealized Waves Cross Section and PPI Filename: Figure3_WaveTilt_xsect_4panel_animation.mp4 DOI: https://doi.org/10.5446/54363

Animation-MotionStudy1: Animated plot of a single wave with a wavelength of 15 km moving toward 305° with a speed of 30 m s$^{-1}$ and a radar volume sampling frequency of 250 s. Title: Wave Motion Study: Sample Speed 250 sec Wave Speed 30

m/s Filename: MotionStudy_ChangingSpeed-ConstantSample_WaveSpeed_30.mp4 DOI:  https://doi.org/10.5446/54386

Animation-MotionStudy2: Animated plot of a single wave with a wavelength of 15 km moving toward 305° with a speed of 40 m s$^{-1}$ and a radar volume sampling frequency of 250 s. Title: Wave Motion Study: Sample Speed 250 sec Wave Speed 40 m/s Filename: MotionStudy_ChangingSpeed-ConstantSample_WaveSpeed_40.mp4 DOI: https://doi.org/10.5446/54387

Animation-Figure-4: Animated plot of aimulated PPIs, RHIs, and vertical cross sections of two superimposed idealized

waves. The longer wavelength wave has a wavelength of 25 km, a depth of 4500 m, a maximum horizontal velocity perturbation of 4 m s$^{-1}$, and moves toward 290° with its long axis along 200°. The shorter wavelength wave has a wavelength of 8 km, a depth of 1000 m, a maximum horizontal velocity perturbation of 2 m s$^{-1}$, and moves toward 250° with its long axis along 160°. Simulated radar has the following characteristics similar to a NWS WSR-88D radar with a beamwidth of 0.95°, 500 m range gates, and maximum range of 150 km. THe upper panels depict simulated radial velocities for a beam elevation of 0.5° and

the corresponding detected velocity waves. The middle panel depicts the cross section of the zonal velocity of the prescribed wind field along the solid gray line in the upper panels. The lower panels depict constructed RHIs of radial velocity and wave detection from a NWS VCP-12-like scan strategy along the same azimuth (90°) as the cross section. The RHI elevations angles match NWS VCP-12. The dashed lines denote the tops of the two idealized waves (goes with Fig. 4). Title: Idealized Waves PPI, Cross Section, and RHI Filename: Figure4_Demo_2_Wave_noBG_animation.mp4 DOI: https://doi.org/10.5446/54364

Animation-Figure-5: Animated plot of reflectivity and dealiased radial velocity for the NWS KOKX precipitation radar in Upton, NY on 1 February 2021 at 9:03 UTC, the difference between the shown radial velocity field and the prior dealiased radial velocity field at 08:56 UTC (not shown), and the resulting binary wave detection field. The red line in each panel denotes the RHI scan from the KASPR radar location at Stony Brook University shown in Figure 6 (goes with Fig. 5). Title: KOKX 20210201 Reflectivity, Velocity, and Waves Filename: KOKX_4panel_PPT_20210201_withKASPR_RHI.mpg.mp4

DOI: https://doi.org/10.5446/54365

Animation-Figure-6: Animated plot of a scanned RHI observed by KASPR radar at Stony Brook University on 1 February 2021 at 09:02:39 UTC along the red lines in Figure 5. RHIs extend from 179° (south) on the left to 359° (north) azimuth on the right. The panels starting at the top are reflectivity, binary wave detection, the difference between the KASPR radar's dealiased radial velocity fields at 9:02:00 UTC and 9:00:41 UTC, and spectrum width field at 9:02 UTC (goes with Fig. 6). Title:

KASPR 20210201 Reflectivity, Waves, Velocity, and Spectrum Width Filename: KASPR_Ref_Waves_DeltaDV_SW_20210201.mp4 DOI: https://doi.org/10.5446/54366





Animation-Figure-7: Animated plot of PPIs and RHIs (b, d, f, h) constructed from PPI scans of reflectivity, radial velocity, the difference of radial velocity for successive scans, and the wave detection field for data from the KBOX WSR-88D radar near Boston, MA at 08:45 UTC on 17 December 2020. The RHIs are constructed from data on the 200° azimuth as denoted

by the radial line on the PPI panels (goes with Fig. 7). Title: KBOX 20201217 PPI and RHI Reflectivity, Velocity, Difference, and Waves Filename: 20201217_KBOX_200_movie.mp4 DOI: https://doi.org/10.5446/54367

Animation-Figure-8: Animated plot of regional radar maps of reflectivity and elongated bands of velocity perturbations detected in radial velocity data for Hurricane Sandy on 30 October 2012 at 03:58 UTC. Data are from twelve NWS radars in the northeast US (goes with Fig. 8). Title: Hurricane Sandy Stitched Reflectivity and Waves Filename: Sandy_Z-Waves_2Panel.mp4

DOI: https://doi.org/10.5446/54368

*Competing interests.* The author's disclose no competing interests.


*Acknowledgements.* Special thanks to McKenzie Peters, Toby Peele, and Levi Lovell for their assistance in data processing. The KASPR observations from the Stony Brook Radar Observatory were provided by Pavlos Kollias and Mariko Oue with support from the National Science Foundation grant AGS-1841246. This work was supported by the National Science Foundation (AGS-1347491 and AGS-1905736) and the National Aeronautics and Space Administration (80NSSC19K0354).





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

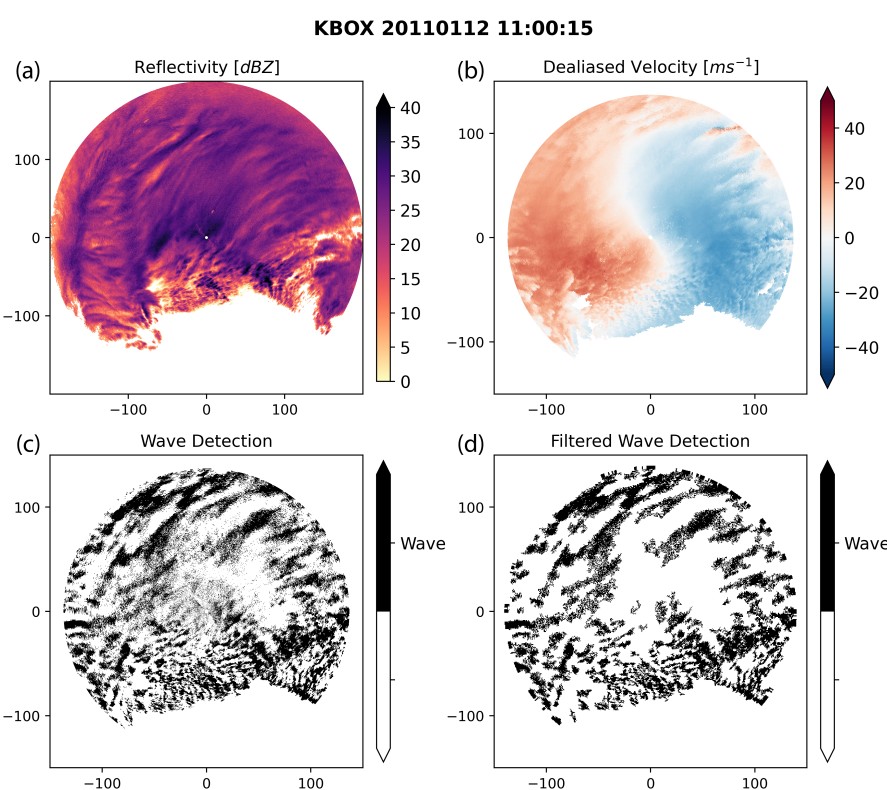

**Figure 1.** Bands of locally enhanced reflectivity and coincident velocity waves from Boston, MA WSR-88D radar (KBOX) on 12 January 2011 at 11:00 UTC. Data shown are 0.5° elevation angle PPIs. (a) Radar reflectivity, (b) dealiased radial velocity, (c) binary version of the difference field between consecutive radial velocity fields all in polar coordinates, (d) Cartesian coordinate (0.5 km grid spacing) filtered version of (c) with eight-connected areas < 16 km$^2$ removed. An animated version of this figure is in the Video Supplement Animation-Figure-1.

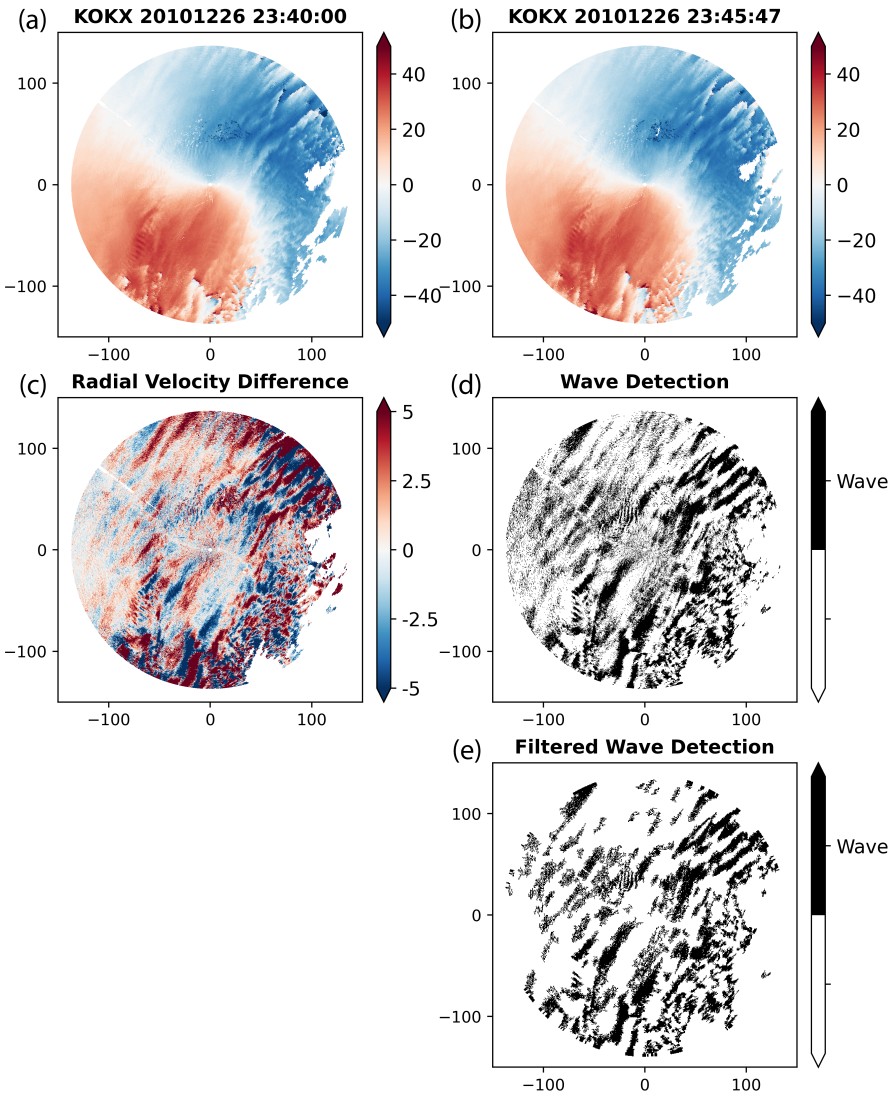

**Figure 2.** Step by step wave detection method illustrated using sequential PPI scans of radial velocity from NWS KOKX radar in Upton, NY on 26 December 2010. Polar coordinate radial velocity fields at 0.5° elevation angle for consecutive scans at (a) 23:40:00 UTC and (b) 23:45:47 UTC. (c) The difference field computed from PPIs (b) minus (a) showing both positive and negative temporal velocity changes. (d) Binary version of the negative portion of the difference field from (c). (e) Cartesian coordinate (0.5 km grid spacing) filtered version of (d) with with eight-connected areas < 16 km$^2$ removed. An animated version of this figure is in the Video Supplement Animation-Figure-2.



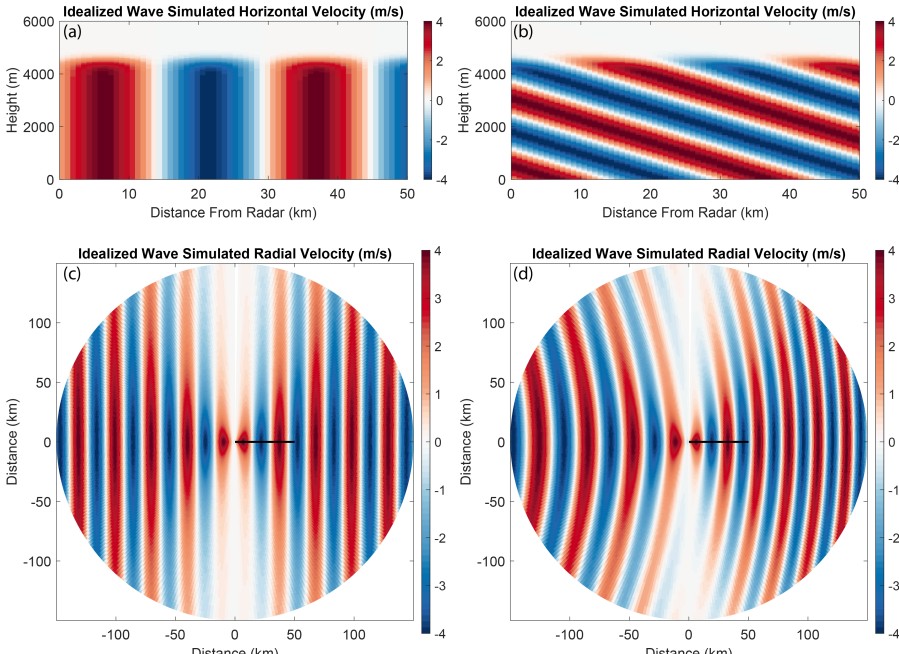

**Figure 3.** Cross sections and simulated radial velocity images of vertical and tilted idealized velocity waves. Both velocity waves have a wavelength of 30.7 km, a depth of 4500 m, and a maximum horizontal velocity perturbation of 4 m s$^{-1}$. The waves in (a) and (c) are upright and the waves in (b) and (d) are tilted. (a) and (b) depict cross sections of the horizontal (zonal) velocity perturbations. (c) and (d) simulate radial velocity observations as would be seen by a NWS WSR-88D precipitation radar with a PPI beam elevation of 0.5°, a maximum range of 150 km, a 0.95° beamwidth, and 500 m range gates. The solid black lines denote the span of the cross sections shown in the upper and middle panels. The vertical and tilted waves have the same cross-band widths. (a) and (b) use a 1:3.8 z-axis to x-axis aspect ratio. An animated version of this figure is in the Video Supplement Animation-Figure-3.

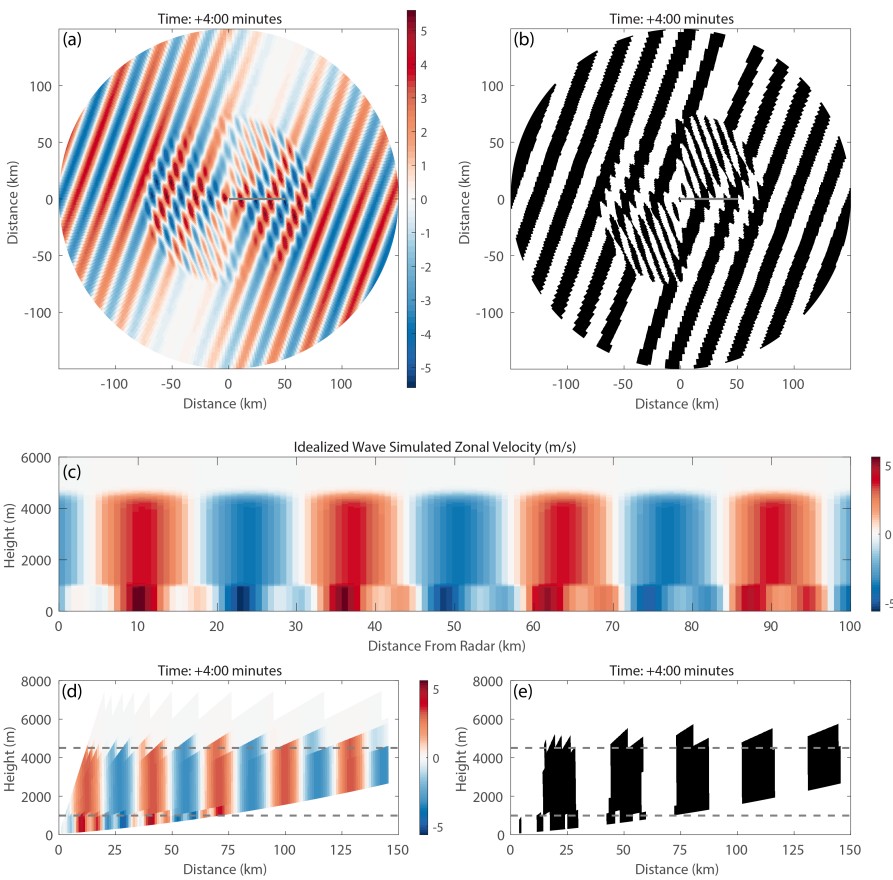

**Figure 4.** Simulated PPIs, RHIs, and vertical cross sections of two superimposed idealized waves. The longer wavelength wave has a wavelength of 25 km, a depth of 4500 m, a maximum horizontal velocity perturbation of 4 m s$^{-1}$, and moves toward 290° with its long axis along 200°. The shorter wavelength wave has a wavelength of 8 km, a depth of 1000 m, a maximum horizontal velocity perturbation of 2 m s$^{-1}$, and moves toward 250° with its long axis along 160°. Simulated radar has the following characteristics similar to a NWS WSR-88D radar beamwidth of 0.95°, 500 m range gates, and maximum range of 150 km. (a) Simulated radial velocities for a beam elevation of 0.5° and (b) the corresponding detected velocity waves. (c) Cross section of the zonal velocity of the prescribed wind field along the solid gray line in the upper panels. (d) Constructed RHI of radial velocity and (e) wave detection from a NWS VCP-12-like scan strategy along the same azimuth (90°) as the cross section in (c). For (d) and (e) elevations angles are 0.5°, 0.9°, 1.3°, 1.8°, 2.4°, 3.1°, 4.0°, 5.1°, 6.4°, 8.0°, 10.0°, 12.0°, 14.0°, 16.7°, and 19.5°. The dashed lines denote the tops of the two idealized waves. (c) uses a 1:3.2 z-axis to x-axis aspect ratio, and (d) and (e) use a 1:8.4 aspect ratio. An animated version of this figure is in Video Supplement Animation-Figure-4.

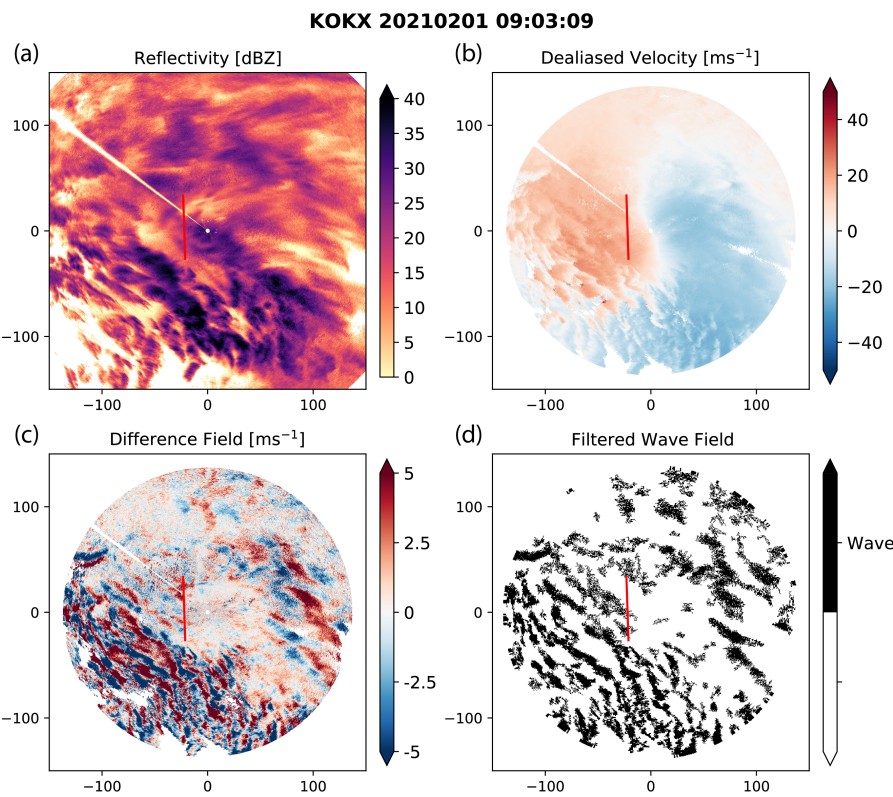

**Figure 5.** (a) Reflectivity and (b) dealiased radial velocity for the NWS KOKX precipitation radar in Upton, NY on 1 February 2021 at 9:03 UTC. (c) The difference between (b) and the prior dealiased radial velocity field at 08:56 UTC (not shown), and (d) the resulting binary wave detection field. The red line in each panel denotes the RHI scan from the KASPR radar location at Stony Brook University shown in Figure 6. An animated version of this figure is in the Video Supplement Animation-Figure-5.

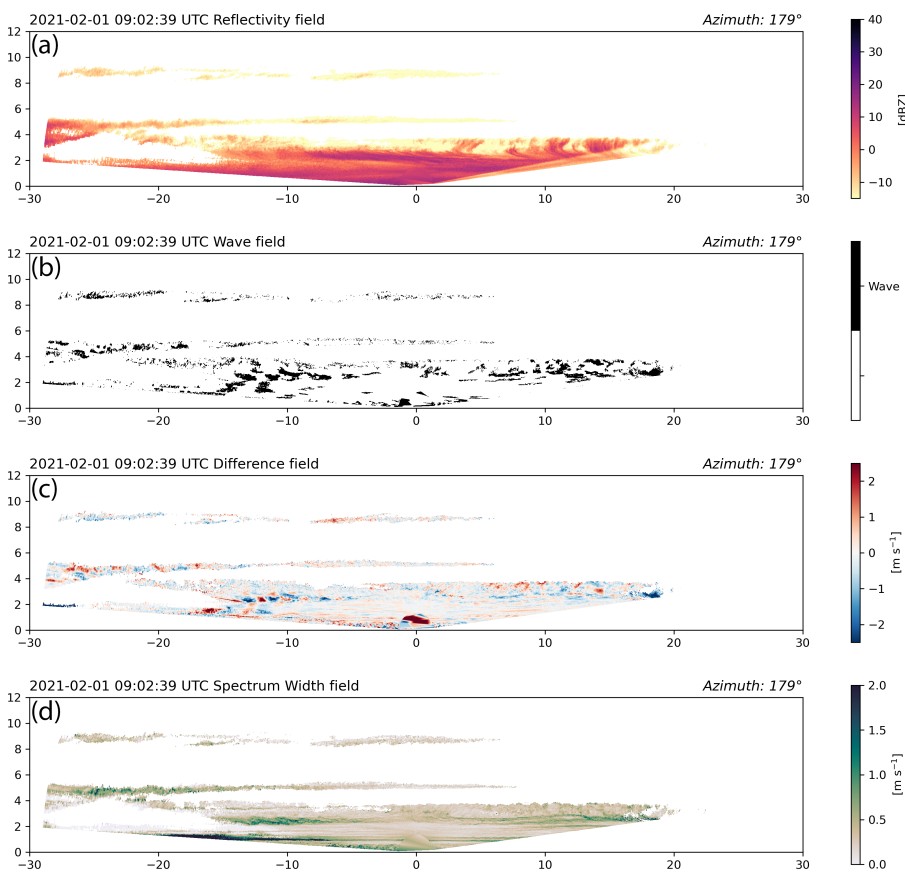

**Figure 6.** Scanned RHI observed by KASPR radar at Stony Brook University on 1 February 2021 at 09:02:39 UTC along the red lines in Figure 5. RHIs extend from 179° (south) on the left to 359° (north) azimuth on the right. (a) Reflectivity, (b) binary wave detection, (c) the difference between the KASPR radar's dealiased radial velocity fields at 9:02:00 UTC and 9:00:41 UTC, and (d) spectrum width field at 9:02 UTC. Images plotted using a 1:1 aspect ratio. An animated version of this figure is in the Video Supplement Animation-Figure-6.

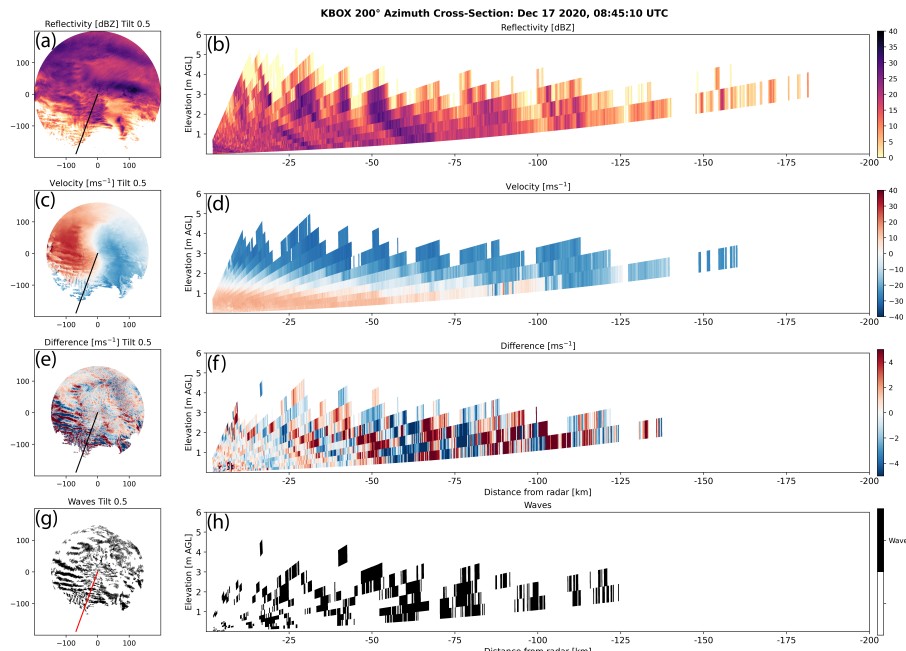

**Figure 7.** PPIs (a, c, e, g) and RHIs (b, d, f, h) constructed from PPI scans of reflectivity, radial velocity, the difference of radial velocity for successive scans, and the wave detection field for data from the KBOX WSR-88D radar near Boston, MA at 08:45 UTC on 17 December 2020. The RHIs are constructed from data on the 200° azimuth as denoted by the radial line on the PPI panels. An animated version of this figure is in the Video Supplement Animation-Figure-7.

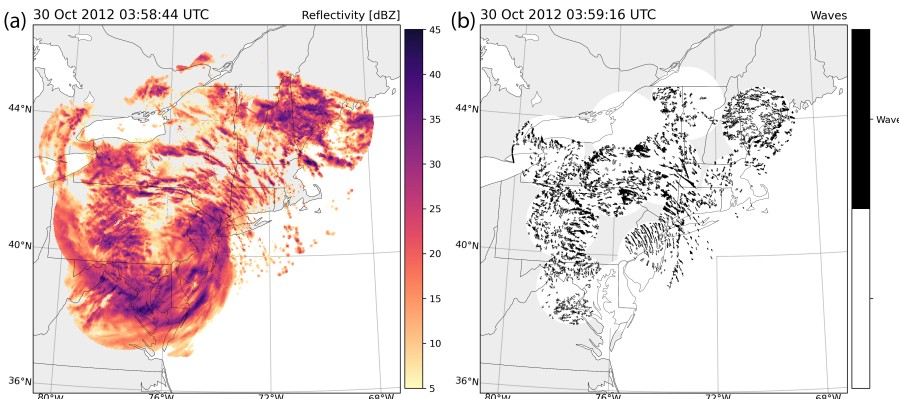

**Figure 8.** (a) Regional radar maps of reflectivity and (b) elongated bands of velocity perturbations detected in radial velocity data for Hurricane Sandy on 30 October 2012 at 03:58 UTC. Data are from twelve NWS radars in the northeast US. An animated version of this figure is in the Video Supplement Animation-Figure-8.