# Peer review of "Detecting Wave Features in Doppler Radial Velocity Radar Observations"

_Atmospheric Measurement Techniques, 2021_

## Author Response (AR1)

We would like to start by thanking the reviewer for volunteering their time and expertise in reviewing our manuscript. We appreciate the constructive feedback. Our responses are made inline to selected reviewer comments below. The original reviewer text is in normal black text. Our replies are indented and in blue.

**Reviewer #1 Comments**

Review of the article titled "Detecting Wave Features in Doppler Radial Velocity Radar Observations" by Miller and coauthors for publication in the journal of atmospheric measurements and techniques.

The authors have presented a new technique for identifying wave-like signatures in the mean Doppler velocity data collected by the WSR-88D S-band radars. The technique relies on subtracting the radial velocity field from two successive PPI scans. The authors have discussed the limitation of the technique in detail using a forward operator simulation. The authors have applied the technique to two cases, one of a winter storm and another one of a extratropical cyclone. The paper is short, straightforward and easy to understand. By their own admission, the author state that the technique cannot be blindly applied to wide set of data, but may prove as a useful starting point for scientists studying severe weather. I recommend the article for publication and mention below few minor things that could be used for improving the article further.

One thing I kept thinking of while reading the paper is the role of collective variability of the hydrometeor fall speed, vertical air motion and horizontal air motion in determining the radial velocity field. Some of the "wave-like" features determined here could be purely referring to changes in hydrometeor fall speed, and large boundary layer eddies that tend to orient themselves along the prevailing wind. This is further complicated by small congestus clouds embedded in the large precipitation field observed by the WSR-88D. It will be good to discuss this issue, and maybe do some back of the envelope calculation of the error such and phenomenon might cause. Thanks.

For the cases we illustrate, the radar elevation angles are sufficiently low that we expect little or no impact on the observed radial velocities. The contribution of the hydrometeor fall speed to the radial velocity is equal to the sine of the radar beam elevation angle. For a 0.5 degree beam, only 0.9% of the fall speed contributes to the radial velocity. Snow with a 1 m/s fall speed contributes only 0.009 m/s and for larger rain or grauple particles with fallspeeds of 5 m/s that is still only a contribution of 0.045 m/s. The radial velocity data for a WSR-88D radar only has a precision of 5 m/s. Thus, we don't expect variations in hydrometeor fall speed to be a factor in the wave detection method.

The WSR-88D radar has insufficient sensitivity to observe non-precipitating clouds such as cumulus congestus.

We have observed boundary layer eddies in research radar data sets from winter storms. These research radar observations typically have a vertical resolution

The authors have only used data from the negative velocity perturbations. It will be good if you can mention if you get similar results if you used positive perturbations. I assume that the positive perturbations will have less contamination from the falling rain, and hence might be more robust.

In practice, we have found that using the negative versus the positive side of the perturbation makes no difference for wave identification. For real-world data where the waves are not perfectly symmetric, the positive and negative parts of the wave do not perfectly mirror each other. We have not noted a difference in noise or anything similar between the positive and negative velocity perturbations. We encourage the reviewer to reexamine the radial velocity difference fields in Figures 2, 5, 6, and 7 along with corresponding binary fields while keeping in mind that the human eye is more sensitive to gradations in red than in blue. Our interpretation of these data both by visual and mathematical comparisons is that the positive and negative areas are not meaningfully different from a wave identification perspective.

The simulator is great. However, I could not understand the fall speed or the drop size distributions used in the Figures 3 and 4. It will be good to mention those. Thanks.

Fall speed and DSD are not considered at all. For the purposes of the idealized simulator we made, the velocity perturbations are prescribed directly.

**Line 27: it will be good to define mesoscale here. I assume you mean meso-beta?**

The bands in question fall on the smaller end of the meso-beta scale. We have updated the manuscript to note the typical length scales in both the short and long axes. The referenced papers collectively document the spectrum of sizes of these bands (both the larger primary bands and the smaller multi-bands) in exhaustive detail.

At what was Line 28 and is now Line 39 we added the sentence, "These bands are typically a few tens of kilometers along the short axis and < 200 km along the long axis."

Line 58: are you results sensitive to the 0 dBz threshold used for removing noise?

It is necessary to dealias the observed radial velocities prior to the wave detection step. We apply the 0 dBZ threshold as a pre-processing step on the WSR-88D data since we found that removing the low-reflectivity-value data greatly improves the results of the automated radial velocity dealiasing function (from the PyART library) and produces a less noisy and easier to interpret radar data field.

If a user wanted to apply this method to cloud radar data where the sensitivity of the radar is better than the S-band WSR-88D radar, filtering out values below 0 dBZ would not be a desired step. We have added a sentence pointing out the threshold value is an instrument/application choice at the end of the first paragraph in the Conclusions section.

At what is now Line 293 we added the sentences, "Specific filtering thresholds for such instruments should be selected consistent with instrument specifications and intended application. Processing of radar observations with different dynamic ranges and noise floors than a WSR-88D radar will likely use different threshold values."

Line 233: It will be good to mention the native resolution of the scanning KaSAPR. Thanks. Less than 200 m sounds vague.

It was not our intention to be vague. Being a research radar, KASPR is often run with different settings. We gave a representative value since the resolution is partially a function of range from the radar since the beam spreads with distance. The resolution is also determined by the length of the range gates which itself is a function of the maximum range of the radar and the number of gates used. We have updated the manuscript to note the half-power beamwidth of 0.32 degrees, the number of range gates (1154), and the length of the range gates (24 m) applicable to the cases we have used in the manuscript.

At what is now Line 242 we altered the sentence to read, "The KASPR radar has a narrow beamwidth (0.32°) and, for this case, a range resolution of 24 m for 1154 range gates yielding a maximum unambiguous range of 29 km.".

Line 272-273: I understand the reasoning behind not classifying the hurricane rain bands as gravity waves, but shouldn't they have a larger wavelength than the mesoscale (~20 km)? This might provide an objective way to discard them.

In this specific case, the length scale of the velocity bands is similar to the winter storm cases. We agree that in some situations that the length scale of the velocity perturbations can be used to constrain what the potential wave types are.

Figure 1, 2, 5, 6 and 8 are missing some of the axis labels.

The figures in question as well as Figure 7 have been updated to add axis labels.

Figure 3 caption mentions upper and middle panels, but the figure only has two rows. Something is missing here.

The caption has been corrected.

The caption for Figure 3 was changed from, "The solid black lines denote the span of the cross sections shown in the upper and middle panels..." to, "...The solid black lines denote the span of the cross sections shown in (a) and (b)...".

**Reviewer #2 Comments**

The manuscript describes a method to detect wave like features in Doppler velocity radar data. The analysis is based on low elevation PPI scans as well as RHI scans. The impressive simulations and the examples show the functionality of the algorithm. The animations provided in the supplement are a great help to understand the situation. The manuscript is well written and deserves publication.

However, and this is my main criticism, I feel slightly unsatisfied since the manuscript describes only the very first (but of course important) step towards (automatic) detection of wave features. The authors do outline this point in the conclusion and have good arguments, but I would feel more convinced from the manuscript if for one of the examples additional steps could be shown. Especially since they admit that in one example wave features in are easier to locate in the difference plots than in the binary detection plot (lines 251 - 255).

So far, we have not found a best way to automate detection. An exploration of how to automate the detection of waves and calculation of velocity band motion will fall down a rabbit hole of exploring image processing techniques. Possible approaches include: 2D wavelets, 2D FFTs, texture analysis, and neural networks. Determining a good enough approach is application and data dependent. Exploring this topic in satisfying detail is beyond the scope of this paper and is a potential candidate for a follow up paper.

**Minor remarks:**

Line 41: how to see vertical motion in low level Doppler data?

Vertical motion is a very small component of radial velocity in low-elevation-scan (0.5 degree elevation angle) Doppler radar data. Vertical motion can be inferred from convergence patterns in the data. Vertical motion observations are not required for wave detection but are needed to confirm the quadrature relationship.

Line 43: somehow this first sentence should be given much earlier. Radar is mentioned already in the beginning of the introduction.

This paragraph was moved to earlier in the paper to what is now Line 23.

Line 50: precipitation fall speed is only relevant when elevation is high. You should mention that Ottersten and Stober use vertical pointing radars.

Stober et al. (2013) made use of the MAARSY radar in Norway which is a vertically oriented phased array radar. They used high elevation angle data and then derived the horizontal wind field using VVP and VAD techniques. The sentences that started on what was Line 44 now start on Line 25 and read, "Stober et al. (2013) used horizontal wind observations derived from high-elevation scans from a vertically oriented phased array radar to detect and measure the properties of gravity waves in the upper atmosphere from 75 to 100~km altitude.".

Ottersten et al. (1973) is a review article which describes the results of several radar studies. Some of the radars described are noted as having "steerable antennas" and the paper include examples of timeheight plots from vertically pointing radars as well as RHI examples from scanning radars. The sentences that started on what was Line 43 now start of Line 23 and read, "For example, Ottersten et al. (1973) describe clear-air radar observations of gravity waves in the troposphere from multiple studies which made use of both scanning and vertically pointing radars." Line 50: it is tradition to outline the structure of the paper at the end of the introduction.

The manuscript has been updated to include such an outline at the end of Section 1 starting at Line 55. The added text is, "Section 2 describes the method to detect waves and outlines several limitations to the method. Section 3 examines examples employing the wave detection on idealized data. Section 4 examines the use of wave detection on data from a winter storm. Section 5 examines a tropical system where the wave detection method produces output that looks like a velocity wave but is not and Section 6 contains our conclusions. There is a video supplement which includes animated versions of all figures in this manuscript".

Line 64: maybe it's time to mention that in radar meteorology positive Doppler velocity are motion away from the radar. People from the lidar world often use the inverse notation. The analysis would also be interesting for long-range (app. 15 km) Doppler lidars.

Section 2, paragraph 3 has been updated to note the radial velocity sign convention typical to precipitation radar. At what was Line 66 and is now Line 74 we added, "Note that the convention for Doppler weather radar is that positive radial velocity values represent motion away from the radar."

We agree with the reviewer that there is a potential for applying the wave detection method to lidar data through we have not made any tests. Section 6 (formerly 4), paragraph 1 was updated to discussing the potential application for lidar data. The new sentences at Line 291 and reads, "This technique should be extendable to Doppler radial velocities from other instruments such as cloud radar and lidar.".

Line 165: mention here that you are using also observations.

The section headings have been changed to make it more clear where idealized data in being used versus real-world data. This is in combination with the added outline paragraph at the end of Section 1.

Line 172: what is the motion speed? I'm confused: if the waves move to the west, doesn't this imply that there is an easterly background field? Or is there an additional external force, which isn't relevant here, moving the system to the west?

Gravity waves can and often do move in directions different from the prevailing wind. For the single idealized wave example, we prescribed the wave motion as moving towards the west. We set the background wind field to zero to make the depiction of the wave very clear.